# A Phase I Dose Escalation Study of Oxaliplatin, Cisplatin and Doxorubicin Applied as PIPAC in Patients with Peritoneal Carcinomatosis

**DOI:** 10.3390/cancers13051060

**Published:** 2021-03-03

**Authors:** Manuela Robella, Michele De Simone, Paola Berchialla, Monica Argenziano, Alice Borsano, Shoeb Ansari, Ornella Abollino, Eleonora Ficiarà, Armando Cinquegrana, Roberta Cavalli, Marco Vaira

**Affiliations:** 1Unit of Surgical Oncology, Candiolo Cancer Institute, FPO—IRCCS, 10060 Candiolo, Italy; michele.desimone@ircc.it (M.D.S.); alice.borsano@ircc.it (A.B.); armando.cinquegrana@ircc.it (A.C.); marco.vaira@ircc.it (M.V.); 2Department of Clinical and Biological Sciences, University of Torino, 10126 Torino, Italy; paola.berchialla@unito.it; 3Department of Drug Science and Technology, University of Torino, 10125 Torino, Italy; monica.argenziano@unito.it (M.A.); shoebanwarmohammedkhawja.ansari@unito.it (S.A.); ornella.abollino@unito.it (O.A.); roberta.cavalli@unito.it (R.C.); 4Department of Neuroscience, University of Torino, 10126 Torino, Italy; eleonora.ficiara@unito.it

**Keywords:** PIPAC, peritoneal carcinomatosis, chemotherapy, phase I, locoregional

## Abstract

**Simple Summary:**

This study is one of the very few phase I trials on intraperitoneal chemotherapy applied as PIPAC. Cisplatin and doxorubicin may be safely used as PIPAC at a dose of 30 mg/m^2^ and 6 mg/m^2^, respectively; oxaliplatin can be used at an intraperitoneal dose of 135 mg/m^2^. No serious adverse event was reported. The dosages achieved to date are the highest ever used in PIPAC. The results of these investigations should be the starting point for further clinical phase II trials regarding repeated PIPAC, possibly associated with systemic chemotherapy.

**Abstract:**

Pressurized Intraperitoneal Aerosol Chemotherapy (PIPAC) is an innovative laparoscopic intraperitoneal chemotherapy approach with the advantage of a deeper tissue penetration. Thus far, oxaliplatin has been administered at an arbitrary dose of 92 mg/m^2^, cisplatin at 7.5 mg/m^2^ and doxorubicin 1.5 mg/m^2^. This is a model-based approach phase I dose escalation study with the aim of identifying the maximum tolerable dose of the three different drugs. The starting dose of oxaliplatin was 100 mg/m^2^; cisplatin was used in association with doxorubicin: 15 mg/m^2^ and 3 mg/m^2^ were the respective starting doses. Safety was assessed according to Common Terminology Criteria for Adverse Events (CTCAE version 4.03). Thirteen patients were submitted to one PIPAC procedure. Seven patients were treated with cisplatin and doxorubicin and 6 patients with oxaliplatin; no dose limiting toxicities and major side effects were found. Common adverse events included postoperative abdominal pain and nausea. The maximum tolerable dose was not reached. The highest dose treated cohort (oxaliplatin 135 mg/m^2^; cisplatin 30 mg/m^2^ and doxorubicin 6 mg/m^2^) tolerated PIPAC well. Serological analyses revealed no trace of doxorubicin at any dose level. Serum levels of cis- and oxaliplatin reached a peak at 60–120 min after PIPAC and were still measurable in the circulation 24 h after the procedure. Cisplatin and doxorubicin may be safely used as PIPAC at a dose of 30 mg/m^2^ and 6 mg/m^2^, respectively; oxaliplatin can be used at an intraperitoneal dose of 135 mg/m^2^. The dosages achieved to date are the highest ever used in PIPAC.

## 1. Introduction

Peritoneal carcinomatosis (PC) is a common synchronous or metachronous occurrence in intra-abdominal malignancies (colo-rectal, ovarian, appendiceal, gastric, pancreatic cancers) or the clinical presentation of primitive peritoneal neoplasms (diffuse peritoneal mesothelioma, primary peritoneal carcinomas). Historically, PC was considered an end-stage pathology in the absence of aggressive therapeutic approaches, usually submitted to systemic chemotherapy and/or debulking surgery with palliative intent. In the last decades, the treatment of this peculiar cancer spread recorded a growth both in interest and technical improvements, drawing new outlines in the management of peritoneal metastases. Curative approach is, unfortunately, still reserved to a minority of selected patients, while most of them are treated with palliative intent. The edge between those two groups of patients is sometimes unclear: new ways to treat these “grey-zone patients” based on low-impact treatment could potentially be the near-future ideal approach to PC.

Although systemic chemotherapy (sCT) is still, nowadays, the cornerstone of PC treatment, in peritoneal-confined disease there is established pharmacokinetic and tumor-related evidence that intraperitoneal chemotherapy (IPC), generally administered through an intraperitoneal-implanted catheter, is advantageous [1]. In fact, IPC is reported to be effective, but is still burdened by pharmacological limitations such as poor drug distribution within the peritoneal cavity and poor penetration into cancer nodules [2,3]; even technical problems like the high complication rate related to the intraperitoneal catheter (infections, obstruction, bleeding, dislocation) are not a negligible limiting factor. With conventional IPC, for the above-mentioned reasons, only 40% of patients are able to complete the expected chemotherapy cycles [4,5].

Pressurized IntraPeritoneal Aerosol Chemotherapy (PIPAC) is a novel IPC concept for the treatment of PC by taking advantage of the physical properties of drug micronization and pressure. In fact, the therapeutic aerosol administered under pressure reported a more homogeneous distribution compared to liquid chemotherapy [6,7], a deeper tissue penetration [8,9] and a higher drug tissue concentration [10]. Promising results have been published in peritoneal metastasis of gastric [11,12,13], ovarian [14,15], colorectal [16], pancreatic [17,18] and hepatobiliary [19] origins, demonstrating that PIPAC is feasible, safe, and well-tolerated, with a good clinical response rate [20,21].

So far, the drugs used in PIPAC include cisplatin (CDDP), doxorubicin (DXR) and oxaliplatin (OXA), but there are few data comparing different dosage schedules: a phase I study in patients with recurrent ovarian cancer reported that PIPAC with CDDP and DXR may be safely used at a dose of 10.5 mg/m^2^ and 2.1 mg/m^2^, respectively, with low systemic toxicity [22]; a recent dose-escalation study described that the recommended phase II dose of PIPAC with OXA is 120 mg/m^2^ [23]. Therefore, we designed a phase I study with three cytotoxic substances: oxaliplatin in PC of intestinal origin and pseudomyxoma peritonei (PMP), and cisplatin plus doxorubicin in peritoneal mesothelioma, ovarian and gastric cancer; aim of the study was to identify dose-limiting toxicities and the maximum-tolerable dose (MTD) of the treatment.

## 2. Materials and Methods

### 2.1. Study Design

This is a prospective phase I single-arm, non-randomized, open label, dose escalation study with cisplatin associated with doxorubicin or oxaliplatin as single agent applied as PIPAC in patients with peritoneal carcinomatosis. The study is part of an amendment of PI-CaP Study (EudraCT Number: 2015-000866-72; ClinicalTrials.gov Identifier NCT02604784). The study was originally designed in 2015 as a single-arm, repeated single-dose trial targeting patients not eligible for systemic chemotherapy. In consideration of the difficulty in enrolling patients not amenable to sCT and the desire to increase the drugs dosages, the study design was subjected to amendment by creating two treatment arms:-Cohort A: patients eligible to receive sCT associated with PIPAC.-Cohort B: patients subjected exclusively to PIPAC procedure according to a dose-escalation design.

This paper will focus on the presentation of the cohort B data concerning the phase I study. The primary endpoint of this study was to define the maximum tolerated dose (MTD) of CDDP + DXR or OXA alone administered once as PIPAC. All participants provided written informed consent prior to undergoing the enrollment. All patients included in the court must not have undergone chemotherapy/major surgery in the last four weeks before the PIPAC procedure. Patients were planned to undergo one PIPAC procedure.

Cisplatin and doxorubicin were used in patients with peritoneal carcinomatosis of ovarian and gastric origin and in primary tumors of the peritoneum. A dose escalation model based on a two agents combination design published by Riviere [24] was adopted. It is an extension of the Continual Reassessment Method in case of two-dimensional dose-escalation, which identifies the MTD of cisplatin and doxorubicin based on the probability of dose limit toxicity (DLT) of each combination of the two agents. A Bayesian logistic model was used as probability model.

The first cohort of participants was treated with cisplatin 15 mg/m^2^ body surface in 150 mL NaCl 0.9% and doxorubicin 3 mg/m^2^ (cohort 1); the following cohort was given cisplatin 30 mg/m^2^ and doxorubicin 6 mg/m^2^ (cohort 2) and the third cohort cisplatin 50 mg/m^2^ and doxorubicin 10 mg/m^2^ (cohort 3). Dose escalation was to be continued by the protocol according to the probability model. The recommended doses of both agents are those of the dose level combination associated with a probability of DLT closes to the DLT probability target at 25%.

Oxaliplatin was used in patients presenting peritoneal carcinomatosis of intestinal origin. An extension of the Continual Reassessment Method based on a two-parameter probability model proposed by Neuenschwander [25] will be used to identify the recommended dose of oxaliplatin.

The first cohort of patients was treated with oxaliplatin 100 mg/m^2^ body surface in 150 dextrose solution 5% (cohort 1); the following cohort was given oxaliplatin 135 mg/m^2^ (cohort 2) and 155 mg/m^2^ for the following one (cohort 3).

### 2.2. Study Population

The study population consisted of patients with inoperable peritoneal mesothelioma and primary peritoneal tumor or unresectable peritoneal metastases from ovarian, gastric, intestinal and appendiceal cancer. All patients were enrolled and treated in Candiolo Cancer Institute, Turin, Italy. Every decision regarding the treatment of these patients was validated by a multidisciplinary team. As a stopping rule, the maximum number of 42 patients is considered.

### 2.3. Inclusion and Exclusion Criteria

Patients eligible for recruitment met ALL of the following criteria: unresectable peritoneal metastasis on peritoneal cytology/histology; age between 18 and 80 years; Eastern Cooperative Oncology Group (ECOG) performance status ≤ 2; adequate liver function (AST/SGOT and/or ALT/SGPT ≤ 2.5× ULN (upper limit of the normal range) or ≤ 5× ULN if liver metastases are present, serum bilirubin ≤ 1.5× ULN); adequate renal function (serum creatinine ≤ 1.5× ULN or creatinine clearance > 50 mL/min); cardiac and pulmonary function preserved; adequate bone marrow function (absolute neutrophil count (ANC) ≥ 1.5 × 10^9^/L, hemoglobin (Hb) ≥ 9 g/dL, platelets (PLT) ≥ 100 × 10^9^/L); total recovery or a CTCAE Grade ≤ 1 from all adverse clinical events of previous chemotherapy, surgery and radiotherapy, except for alopecia; no chemotherapy/major surgery in the last four weeks prior to the PIPAC procedure; informed written consent signed.

Any of the following was considered an exclusion criterion: extra abdominal metastatic disease (with the exception of isolated pleural carcinomatosis); bowel obstruction; history of allergic reactions to cisplatin/doxorubicin/oxaliplatin or their derivatives; severe renal failure, myelosuppression, severe hepatic failure, severe heart failure, recent myocardial infarction, severe arrhythmia; immunosuppressed patients, undergoing immunosuppressive therapy; previous treatment with reaching the maximum cumulative dose of doxorubicin, daunorubicin, epirubicin, idarubicin and/or other anthracyclines and anthracenedions; pregnancy; patients of both sexes with reproductive potential who refuse to use an adequate means of contraception; major surgery or chemotherapy less than four weeks prior to PIPAC procedure.

### 2.4. Treatment

Enrollment clinical examination included: standard physical examination; hematologic, liver and renal function tests; urinalysis; cardiopulmonary function evaluation tests; thorax-abdomen contrast-enhanced computed tomography (CE CT) performed no more than 4 weeks before PIPAC. Doxorubicin, cisplatin and oxaliplatin plasma levels were assayed with blood samples drawn prior to and 30, 60, 120 min, 6 h, 12 h and 24 h after PIPAC procedure at each dose level. Drug tissue levels were dosed with multiple biopsies before and after each PIPAC.

Patients were recruited consecutively. Venous thromboembolism prophylaxis was given the night before surgery using Enoxaparin 4000 IE; a single intravenous injection of cefazolin 2 g was administered about 30 min prior to surgery. PIPAC procedures were performed as previously described [26]. Under general anesthesia, a midline single-port access (Olympus Quadport + platform) was performed and a capnoperitoneum of 12 mm Hg at 37 °C was applied [27]. Explorative laparoscopy was performed, and Peritoneal Cancer Index (PCI) was determined. Peritoneal biopsies were taken for histological examination and baseline tissue drug concentration detection. Ascites was aspirated, measured and sent for cytologic examination. A nebulizer (Capnopen^®^, Reger Medizintechnik, GmbH, Villingendorf, Germany) was then connected to an intravenous high-pressure injector (Arterion Mark 7^®^, Bayer HealthCare, MEDRAD Europe B.V., Beek, The Netherlands) and inserted into the abdomen. The tightness of the abdomen was controlled through a zero flow of CO_2_. A pressurized aerosol containing oxaliplatin at the respective dose according to the dose escalation design in 150 mL dextrose solution was applied in patients with intestinal peritoneal carcinomatosis; an aerosol containing cisplatin at the respective dose in 150 mL NaCl 0.9% and doxorubicin at the respective dose in 50 mL NaCl 0.9% was administered in patients with gastric, ovarian and primary peritoneal cancer. The chemotherapy injections were remote-controlled, and no personnel remained in the operating room during the application. The flow rate was 30 mL/min and the maximal upstream pressure was 200 psi. The system was maintained for 30 min, and then exsufflated via a closed line over two sequential micro-particle filters into the air waste system of the hospital. Peritoneal samples were taken for drug tissue concentration evaluation. The single-port access was removed. No abdominal drainage was applied.

### 2.5. Post-Operative Follow-Up

Patients were followed for 28 days after PIPAC procedure; hematologic exams (including blood and electrolytic counts, liver and renal function parameters) were performed on post-operative day (POD) 0, 1, 2, 15, 28. Between 4 and 6 weeks after the procedure, a CE CT scan of thorax and abdomen was done (anticipated in case of hematologic or clinical alterations). Doxorubicin, cisplatin and oxaliplatin plasma levels were assayed with blood samples drawn prior to and 30, 60, 120 min, 6 h, 12 h and 24 h after PIPAC procedure at each dose level.

### 2.6. Pharmacokinetics and Pharmacodynamics

#### 2.6.1. HPLC (High Performance Liquid Chromatography) Quantitative Determination of Doxorubicin

Quantitative determination of doxorubicin was performed by HPLC analysis [26]. A HPLC system consisting of a pump (Shimadzu LC-9A PUMP C) equipped with fluorescence detector (Chrompack) was used. The mobile phase was a mixture of KH_2_PO_4_ (0.01 M, pH 1.4), acetonitrile and methanol (65:25:10 *v*/*v*/*v*), degassed and pumped through the column (Agilent TC C18 column, 250 mm × 4.6 mm, 5 µm) with 1 mL/min flow rate. Excitation and emission wavelengths of 480 and 560 nm, respectively, were set to monitor the column effluent and determine doxorubicin concentration. For calculating the drug concentration, the external standard procedure was used. A stock standard solution was obtained by weighing an amount of doxorubicin hydrochloride (about 1 mg) in a volumetric flask and dissolving it using filtered water. The stock standard solution was then diluted with the mobile phase, to prepare a series of calibration solutions. Subsequently, the standard solutions were injected into the HPLC system. The calibration curve was linear over the concentration range between 5–100 ng/mL, showing a regression coefficient of 0.999.

#### 2.6.2. Preparation of Plasma Samples for Doxorubicin Analysis

For doxorubicin quantitative determination all plasma samples were treated following a previously tuned procedure. A volume of plasma was mixed with two volumes of methanol, to precipitate plasma protein by means of vortex. Then, precipitated proteins were removed by centrifugation at 10,000× *g* for 5 min and 100 µL of the clear supernatants were injected after suitable dilution for HPLC analysis. The doxorubicin concentration in plasma samples was calculated using a doxorubicin calibration curve in plasma. More specifically, each doxorubicin plasma standard was prepared by spiking blank plasma with increasing concentrations of the drug

#### 2.6.3. Quantitative Determination of Platinum in Plasma Samples

To determine platinum concentration in plasma, ICP–MS measurements were carried out by means of a high-resolution ICP–MS (Element 2, Thermo Finnigan, Bremen, Germany) [28]. The instrument is equipped with a torch using a Pt-guard electrode. Sample transport to the nebulizers was established by using a peristaltic pump. The up-take time of samples was 40 s approximately 800 μL/min, and the sampling time was set at 2 min. All measurements were carried out under clean room conditions with filtered, temperature-controlled, and excess pressured air.

For the ICP-MS analysis, the plasma samples underwent a digestion procedure that involved heating the samples in the presence of mineral acid such as nitric acid and other oxidizing agents such as hydrogen peroxide, to destroy completely plasma proteins. For this purpose, 500 µL of plasma sample were weighed in the digestion vials and 1 mL of concentrated nitric acid and 1 mL of hydrogen peroxide were added. The mixture was initially kept at room temperature for 30 min and then the digestor block temperature was slightly increased to 40 °C for 30 min and finally up to 80 °C for 3 h. After cooling at room temperature, the samples were filled up to 20 mL with ultrapure water for subsequent analysis.

### 2.7. Outcome Measures

The primary endpoint of the study was the incidence of dose-limiting toxicity. Toxicity was graded using the National Cancer Institute (NCI) Common Terminology Criteria for Adverse Events (CTCAE) version 4.03. Patients were assessed on day 0, 1, 2 (until the date of discharge), 15, 28 for toxicities, adverse events, hematology and chemistries.

Secondary endpoints were: assessment of serum concentration of drugs after PIPAC procedure; evaluation of the clinical tumor response based on RECIST criteria (version 1.1) after PIPAC.

### 2.8. Statistical Analysis

Descriptive statistics summarized patient demographics, AEs, clinical laboratory evaluations and safety data of cisplatin, doxorubicin and oxaliplatin administered as PIPAC. Continuous variables were reported as mean and range and categorical variables as absolute frequencies and percentage.

## 3. Results

Thirteen patients were enrolled in the study from May 2011 to November 2019. Patient demographic and perioperative features are listed in Table 1.

A cohort size of three patients was planned. Overall, considering the population treated with cisplatin and doxorubicin, three patients were enrolled in cohort 1, three patients in cohort 2 and one patient in cohort 3. For the oxaliplatin-treated population, two cohorts of three patients were enrolled. Enrollment was stopped due to the term of insurance coverage and bureaucratic issues relating to the renewal. No dose limiting toxicities were found. The recommended dose for cisplatin and doxorubicin administered as PIPAC was therefore established, based on the escalation foreseen by the probabilistic model, at 30 mg/m^2^ and 6 mg/m^2^, respectively; the recommended dose for oxaliplatin was 135 mg/m^2^.

There were no perioperative deaths. Common adverse events included abdominal pain and nausea, mostly mild and moderate, unrelated to drug dose. Abdominal pain was found to be a more frequent complication in patients treated with oxaliplatin. All AEs are presented in Table 2 and Table 3.

Liver, renal and hematologic toxicities were not observed in any cohort. A transient rise in the leucocyte counts was noted during POD 1–2 in almost all patients, with subsequent rapid return to baseline values. Mean hospital stay was 2.6 nights (range 2–7); the third patient of cohort 2 treated with CDDP and DXR most reported complications due to the advanced state of disease and the rapid cancer progression: she complained of nausea, emesis, and ileus and was discharged after 7 days with a nasogastric tube and total parenteral nutrition.

The CE CT scan performed 4–6 weeks after the PIPAC procedure reported no intra-abdominal collections or abscess; disease stability according to RECIST Criteria 1.1 was registered in 12 patients. One patient in cohort 2 treated with CDDP and DXR died 41 days after PIPAC due to progressive disease.

Pharmacokinetics studies reported no trace of DXR at any dose level. Maximum platin concentration in plasma was observed 60 min after PIPAC in cohort 1 and 2, with a second peak at 12 h in the second dose level. In the first group, the serum level of CDDP remained stable over time and was still measurable after 24 h; in the second group, platinum concentration was found to decrease after the peak at 12 h, while still remaining measurable after 24 h. Data of third cohort patient are incomplete due to sample deterioration. Similar results were reported in patients treated with oxaliplatin: the plasma concentration of the drug reached a plasma peak 30 min after PIPAC and drug levels were still detectable 24 h after the procedure. These results could be related to the interaction of the two platin compounds with plasma albumin. Figure 1 and Figure 2 show the respective serum levels of CDDP and OXA prior to and 30 min, 60 min, 120 min, 6 h, 12 h and 24 h after PIPAC.

## 4. Discussion

Peritoneal carcinomatosis is a common and challenging evolution of several tumors, including gastro-intestinal, ovarian, hepatobiliary and pancreatic cancer; moreover, it could be the clinical presentation of primitive peritoneal neoplasms such as diffuse peritoneal mesothelioma and primary peritoneal carcinomas. The therapy of peritoneal carcinomatosis is largely palliative, based on systemic chemotherapy: its low efficacy is due to the poor peritoneal drug distribution and penetration into cancer nodules [29,30]. A significant pharmacological advantage is given by intraperitoneal chemotherapy, burdened anyhow by a large number of unfinished treatments for complications related to the intraperitoneal catheter. In this scenario, the PIPAC procedure added the advantage of pressurized aerosolization and drug micronization, achieving better distribution and penetration into peritoneal tissues with a negligible systemic toxicity [6,10].

Evidence from retrospective and prospective studies demonstrated that the arbitrarily established doses applied as PIPAC of oxaliplatin 92 mg/m^2^ and doxorubicin 1.5 mg/m^2^ associated with cisplatin 7.5 mg/m^2^ are safe, well tolerated and efficacious [31].

To our knowledge there is only one published phase I trial evaluating the use of cisplatin and doxorubicin applied as pressurized intraperitoneal aerosol in patients with PC of ovarian origin. The dose escalation design was based on very low dose increases between steps: the maximum tolerable dose was not reached, and the study was stopped after three dose steps, suggesting that PIPAC with CDDP and DXR may be safely used at an IP dose of 10.5 mg/m^2^ and 2.1 mg/m^2^, respectively [22]. If there are qualms concerning PIPAC related to low drugs doses, this study does not resolve these questions, as a result of using a recommended dose that is still too low.

Similarly to PIPAC with CDDP and DXR, the administration of oxaliplatin as pressurized intraperitoneal aerosol was also the subject of only one phase I study, reporting that the recommended phase II dose should be 120 mg/m^2^ [23].

On the basis of the cited literature, this study aimed to increase the scientific evidence regarding the dosages used: in particular, the objective was to search for the MTD of the drugs applied in a single PIPAC procedure.

The choice to perform a single PIPAC administration was based on two considerations: the number of PIPACs that can be performed in a patient is variable and the possibility of performing further procedures often decreases over time. This is evident in both phase I trials: in the study by Tempfer [22], eight patients (53.3%) were submitted to three PIPAC procedures, three patients (20%) to two PIPACs and four patients (26.7%) to one PIPAC; this aspect is even more evident in the study by Kim [23], in which 50% of the patients underwent only one PIPAC procedure.

If the multiple retrospective and prospective studies published to date have shown the absence of cumulative toxicity with PIPAC, the possibility of using higher dosages would in any case not exclude the chance of repeated treatments, extending the time intervals, if necessary, between procedures.

This study consisted of a prospective, phase I, model-based approach for dose escalation design with significant dose increases between one level and the following one, but with the possibility, in the case of toxicity, of testing intermediate doses. Our results show that CDDP and DXR can be safely administered as PIPAC at a dose of 30 mg/m^2^ and 6 mg/m^2^, while OXA can be used at a dose of 135 mg/m^2^.

As reported in the two phase I studies [22,23], we did not register any liver and renal function alteration and no systemic hematologic toxicity was observed.

In the PIPAC-OX trial, three cases of acute pancreatitis (of which one G3 complication) are reported; in our case history, neither analogous cases nor cases of serum amylase and/or lipase elevation were recorded [23]. As reported by the authors themselves, acute pancreatitis is not a complication typically related to the administration of OXA as PIPAC. On the contrary, abdominal pain, which in our series of patients treated with OXA turned out to be the most frequent complication, is known to be related to the intraperitoneal administration of this drug [32].

Analyzing our postoperative morbidity in detail, one patient in the second dose level treated with CDDP and DXR complained of a severe pain localized on the surgical wound; this resolved, however, with only paracetamol. It was considered a surgical complication, and not a result of high drug doses. In the same subgroup, patient n. 3 reported several postoperative complications (nausea, emesis, ileus and anemia) and passed away 41 days after discharge. Her postoperative morbidity was attributed to the advanced stage of the disease and to the particularly poor general conditions; considering the young age of the patient and the lack of therapeutic alternatives, her enrollment was multidisciplinarily approved. The need to enroll a further triplet of patients on the basis of her postoperative complications was assessed jointly with all of the co-investigators: it was therefore decided to move on to the next dose level.

In terms of efficacy based on RECIST criteria evaluation, considering patients submitted to one PIPAC procedure, PIPAC-OX study reported a disease stability in 62.5% of patients, compared to 92.3% (12/13) in our analysis. Although the TC scan is notoriously limited for the evaluation of peritoneal carcinomatosis (especially of the mesenteric diffusion, present in most of the patients included in the study), the results obtained are encouraging, and it can be improved with the association with systemic chemotherapy.

Similarly to what was reported in the Singaporean study [23], the systemic concentration of oxaliplatin increases exponentially in the different dose levels: the maximum serum OXA concentration in our highest cohort treated with 135 mg/m^2^ resulted the 30% of that reported for intravenous infusion of oxaliplatin 130 mg/m^2^ (1146 ng/mL vs. 2.59–3.22 μg/mL) [33]. As reported in the literature, no significant accumulation was observed in plasma ultrafiltrate after multiple dosing at 130 mg/m^2^ every 3 weeks [33,34]: it can be deduced that even after multiple PIPAC procedures, no plasma accumulation of the drug should be detected.

Regarding cisplatin, we observed a platinum peak of 1494 ng/mL in the third dose level: we can compare it to data from literature where, with a dosage of 20 mg/m^2^/day over five days, a mean CDDP serum peak level of 1.89 μg/mL is reported [35]. Furthermore, as reported in the literature, the highest concentrations observed for CDDP are lower than the ones obtained after intravenous injection of cisplatin at 100 mg/m^2^ [36]. The pharmacokinetics results highlighted a very slow decrease of the plasma concentration of CDDP and OXA over time. This behavior can be related to the binding of the two platin compounds to the plasma albumin [37,38].

These analyses suggested the chance of repeating the PIPAC procedure safely and combining PIPAC with systemic chemotherapy with acceptable systemic toxicities.

An important limitation of the study is the early term of patient enrollment due to the lack of insurance coverage. Despite this, the dosages safely used are the highest adopted up to now: cisplatin and doxorubicin may be used as PIPAC at a dose of 30 mg/m^2^ and 6 mg/m^2^, respectively; oxaliplatin can be used at an intraperitoneal dose of 135 mg/m^2^. As it was planned for patients to undergo only one PIPAC procedure, although as already mentioned no plasma accumulation has been reported in the literature for intravenous application, any systemic complications such as neuropathy or cumulative toxicity due to repeated PIPAC such as bowel sclerosis [22] were not evaluated.

## 5. Conclusions

This study demonstrated the feasibility and safety of oxaliplatin, cisplatin and doxorubicin administered as pressurized intraperitoneal aerosol at higher doses than the standard ones. The results reported herein are encouraging, considering the toxicity of chemotherapeutic drugs.

The data suggest that oxaliplatin can be used safely as PIPAC at an intraperitoneal dose of 135 mg/m^2^. A French study about PIPAC associated with systemic chemotherapy reported dose limiting toxicities at the dose of 140 mg/m^2^, concluding that the recommended OXA dose should be 90 mg/m^2^ [39]. Furthermore, the Singaporean study reported an improvement in PRGS scores and PCI at all dose cohorts tested from 45 mg/m^2^ to 120 mg/m^2^; a dose-proportional efficacy of oxaliplatin is evident [23]. For these reasons, the intermediate dose of 135 mg/m^2^ herein investigated may be the one that benefits most from dose escalation, without dose limiting toxicities.

Cisplatin and doxorubicin may be safely used as PIPAC at a dose of 30 mg/m^2^ and 6 mg/m^2^, respectively. The combination of CDDP and DXR appears to be one of the most effective available regimens with acceptable locoregional toxicity.

No DXR trace was found in serum. Serum platin compounds reached a peak 30 min after PIPAC treatment, followed by a slow decrease over time, probably related to their binding with plasma albumin. Our results suggest that exposure to platin compounds will increase proportionally and predictably with PIPAC doses; considering their favorable pharmacokinetics, it is possible to plan multiple administrations of PIPAC, in the case of delaying the procedures.

These results should not inspire an increase in the use of high-doses PIPAC outside of clinical trials. Considering the type of study, aimed at identifying the MTD, we do not have data in terms of therapeutic effects. These results could be the starting point for future phase II studies in order to evaluate the efficacy of repeated high-doses PIPAC, possibly associated with systemic chemotherapy.

## Figures and Tables

**Figure 1 cancers-13-01060-f001:**
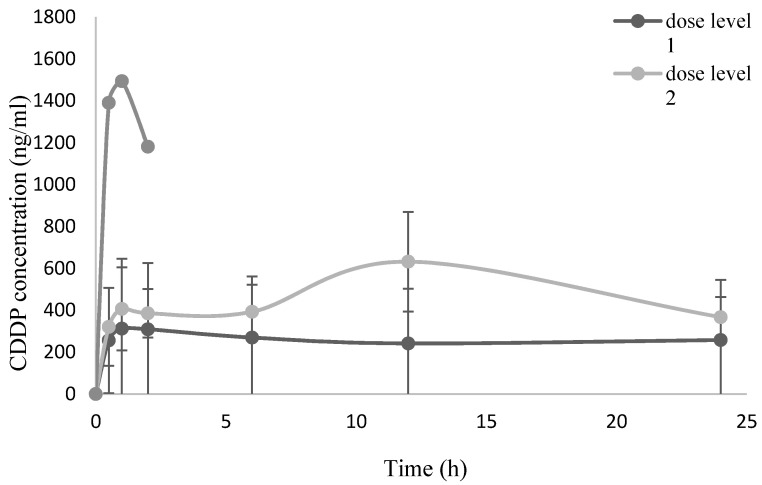
CDDP plasma concentration (ng/mL) over time after PIPAC procedure. Results are expressed as mean ± SD.

**Figure 2 cancers-13-01060-f002:**
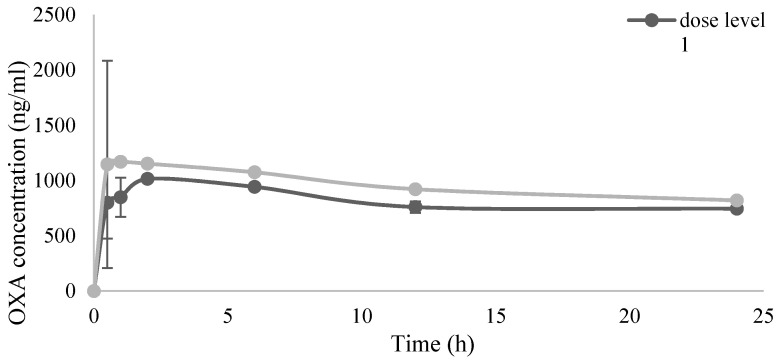
OXA plasma concentration (ng/mL) over time after PIPAC procedure. Results are expressed as mean ± SD.

**Table 1 cancers-13-01060-t001:** Demographic clinical and perioperative features of patients.

Variable	*n* = 13
Age (y), mean (range)	62.2 (34–79)
Females	9 (69%)
ECOG Performance Status	
0	5 (38%)
1	7 (54%)
2	1 (8%)
ASA Score	
1	0 (0%)
2	7 (54%)
3	6 (46%)
Body Surface Area, mean (range)	1.74 (1.32–2.12)
Histology	
EOC	2 (16%)
CRC	5 (38%)
GC	5 (38%)
PMP	1 (8%)
Prior Surgical Score	
1	5 (38%)
2	5 (38%)
3	3 (24%)
PCI, mean (range)	14 (6–24)
Ascites	
No	8 (62%)
Yes	5 (38%)
0–500 mL	4 (32%)
>500 mL	1 (8%)
Operative time (min), mean (range)	91 (55–125)

EOC = epithelial ovarian cancer; CRC = colorectal cancer; GC = gastric cancer; PMP = pseudomyxoma peritonei.

**Table 2 cancers-13-01060-t002:** Adverse events (CTCAE 4.03) according to dose level of Cisplatin and Doxorubicin.

Adverse Event	CDDP 15 mg/m^2^ + DXR 3 mg/m^2^			CDDP 30 mg/m^2^ + DXR 6 mg/m^2^			CDDP 50 mg/m^2^ + DXR 10 mg/m^2^
	Pt 1	Pt 2	Pt 3	Pt 1	Pt 2	Pt 3	Pt 1
Pain					3	2	
Nausea						3	1
Emesis						3	
Ileus						3	
Anemia						3	
Hypokalemia						1	

**Table 3 cancers-13-01060-t003:** Adverse events (CTCAE 4.03) according to dose level of Oxaliplatin.

Adverse Event	OXA 100 mg/m^2^			OXA 135 mg/m^2^		
	Pt 1	Pt 2	Pt 3	Pt 1	Pt 2	Pt 3
Pain	1		2	1	2	1
Nausea						1
INR increased		1				

## Data Availability

The data presented in this study are available on request from the corresponding author. The data are not publicly available due to privacy reasons.

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
