# Peer review of "A Phase I Dose Escalation Study of Oxaliplatin, Cisplatin and Doxorubicin Applied as PIPAC in Patients with Peritoneal Carcinomatosis"

_cancers, 2021, doi:10.3390/cancers13051060_

Round 1

Reviewer 1 Report

The aim of the study was to identify the MTD for cisplatin/doxorubicin, and oxaliplatin administered using novel PIPAC delivery system. While the study was well designed and the manuscript was well written, the major limitation of the study was that MTD was not attained because the study was terminated early due to 'insurance coverage and bureaucratic issues'.

Additional major concerns:

  1. 2 of 3 in CDDP/DXR cohort had grade 3 toxicities. Toxicities in patient 3 were related to disease progression. Author did not explain why G3 pain after PIPAC was not a DLT. Detail description and management for the G3 pain was not provided. Typically the presence of a G3 toxicity in 1 of 3 subjects would entail an expansion to additional 3 more subjects in a 3+3 design.
  2. No information on the median cycle of treatment, hence tolerability of repeated PIPAC at higher doses cannot be determined. 
  3. It is a missed opportunity that pathological response following PIPAC was provided or evaluated. If study subjects had received more than 1 PIPAC, evaluation of pathological response may shed light on the potential effect of a higher dose PIPAC compared with the lower dose cohort.
  4. An internal standard is required for bio-analysis of oncology drug such as doxorubicin.
  5. Plasma filtrate platinum should be measured instead of serum concentration because it represents the active free drugs. In addition, oxaliplatin and cisplatin have different albumin binding properties.

Minor comments:

  1. Grade 2 pain were observed in OXA 100mg/m2 and 135mg/m2 cohort. Pancreatitis had been reported in OXA PIPAC. Author should comment if amylase or lipase were elevated and the management of the pain. 
  2. mg/sm should be replaced with mg/m2 

Author Response

REVIEWER 1

  1.  2 of 3 in CDDP/DXR cohort had grade 3 toxicities. Toxicities in patient 3 were related to disease progression. Author did not explain why G3 pain after PIPAC was not a DLT. Detail description and management for the G3 pain was not provided. Typically the presence of a G3 toxicity in 1 of 3 subjects would entail an expansion to additional 3 more subjects in a 3+3 design.

Thank you for the remark. Patient n. 2 presented a severe pain localized on the surgical wound (grade 3 according to CTCAE 4.03 = a disorder characterized by a sensation of marked discomfort limiting self-care ADL), resolved, however, with only paracetamol. It was considered a surgical complication, not due to high drugs doses. On the other hand, considering the adverse events that occurred in patient n. 3 as partly due to the advanced stage of the disease and to the particularly poor general conditions, the need to enroll a further triplet of patients was assessed jointly with all the co-investigators: it was therefore decided to move on to the next dose level.

  1. No information on the median cycle of treatment, hence tolerability of repeated PIPAC at higher doses cannot be determined. 

Thank you for the consideration. As reported in the paper, patients were planned to undergo only one PIPAC procedure: systemic complications due to repeated PIPAC cannot be evaluated and this aspect represents a limitation of the study. The rationale for this choice lies in the consideration that the possibility of performing repeated procedures often decreases over time. In fact, in the two phase 1 PIPAC studies published so far, patients were planned to undergo at least 2 PIPAC procedures, but due to general conditions deterioration or intraperitoneal adhesions, in the study by Tempfer, only 53.3% were submitted to 3 PIPACs and in the study by Kim the 50% of patients underwent only one PIPAC procedure. 

  1. It is a missed opportunity that pathological response following PIPAC was provided or evaluated. If study subjects had received more than 1 PIPAC, evaluation of pathological response may shed light on the potential effect of a higher dose PIPAC compared with the lower dose cohort.

The main objective of the study was to investigate the MTD of PIPAC, not to evaluate its efficacy. Despite this, the CE CT scan performed 4-6 weeks after PIPAC procedure reported a disease stability according to RECIST Criteria 1.1 in 12 out of 13 patients. Considering the advanced stage of disease of these patients and the possibility to treat them only with PIPAC, the promising results in terms of radiological response should be the starting point for furthers phase 2 studies aimed at evaluating its efficacy in terms of radiological and histopathological response. 

  1. An internal standard is required for bio-analysis of oncology drug such as doxorubicin.

Thank you for the comment. We didn’t use an internal standard as a preliminary experiment has shown that the plasma standard preparation method herein used was able to completely recover doxorubicin. The doxorubicin concentration in plasma samples was calculated using a calibration curve of doxorubicin in plasma. More in particular, each doxorubicin plasma standard was prepared by spiking blank plasma with increasing concentrations of the drug. The protein precipitation method was adopted for both plasma standards and plasma samples. The Material and Methods section was modified accordingly.

  1. Plasma filtrate platinum should be measured instead of serum concentration because it represents the active free drugs. In addition, oxaliplatin and cisplatin have different albumin binding properties.

We agree with the Reviewer comment. Our objective was to determine the total amount of oxaliplatin and cisplatin in plasma after PIPAC treatment.

Minor comments:

  1. Grade 2 pain were observed in OXA 100mg/m2 and 135mg/m2 cohort. Pancreatitis had been reported in OXA PIPAC. Author should comment if amylase or lipase were elevated and the management of the pain. 

Thank you for the remark. In our experience the administration of oxaliplatin, even at standard doses, has always been more painful than PIPAC with CDDP and DXR. This aspect was also evident in this dose-escalating study: pain was reported in 2/7 (29%) patients treated with cisplatin and doxorubicin and in 5/6 (83%) patients treated with oxaliplatin. However, the painful symptoms were not related to the presence of pancreatitis: in fact, in none of our patients amylase and lipase increased in the postoperative period.

Considering the management of the pain, it was generally well controlled with paracetamol at fixed times and tramadol as needed in very rare cases.

  1. mg/sm should be replaced with mg/m2 

Thank you for the note, it has been revised as requested

Reviewer 2 Report

The authors report a phase I study about different drugs applied during PIPAC procedure for peritoneal carcinomatosis from different primary.

PIPAC is an innovative therapeutic, currently in development. The studies of safety and dose escalading are importants. The results of this study are interesting and could change the daily practice because the maximal tolerable dose are more important than currently dose used.

My comments are

Major Comment

  • The method of dose escalading is unclear. The authors describe a continual reassessment method in the chapiter “method” and describe another method of dose escalading, the 3 X 3 method in “abstract” “results” and “discussion” chapiter. The method is confusing and has to be clarify. Minor comment
  •  
  • In method, the authors described a PK analysis at 60, 120 min, 6 h, 12 h after PIPAC but results at 24 hours are available. Is it a mistake ?
  •  
  • The PK analysis showed no or very slow plasma decrease of CDDP and OXA. In figures, the plasma concentration at 24 hours is close to concentration at 1 hour. Theses results are surprising and could be discussed.

  • The authors describe in the chapiter « method » a drug tissue analysis with multiple biopsies. However no data are available in results.
  •  
  • The authors discuss the cumulative toxicity. It’s relevant for repeat cycle of chemotherapy. However, no clinical or pharmaokinetic data are avalaible in this study to assess this cumulative toxicity. The analysis was done after the first PIPAC and not after the others. Some data of number of PIPAC done and toxicities after second or third PIPAC could be interesting to assess the cumulative toxicity.
  • Another phase I study of PIPAC with oxaliplatin found a MTD of 90mg/m2, lower than 135mg/m2 of this study. The authors could discuss this differents results.

Author Response

REVIEWER 2

The authors report a phase I study about different drugs applied during PIPAC procedure for peritoneal carcinomatosis from different primary.

PIPAC is an innovative therapeutic, currently in development. The studies of safety and dose escalading are important. The results of this study are interesting and could change the daily practice because the maximal tolerable dose are more important than currently dose used.

My comments are

Major Comment

  • The method of dose escalading is unclear. The authors describe a continual reassessment method in the chapter “method” and describe another method of dose escalading, the 3 X 3 method in “abstract” “results” and “discussion” chapter. The method is confusing and has to be clarify.

We thank the reviewer for pointing out this issue. The design is a model approach and we have now corrected the confusing description.

Minor comment

  • In method, the authors described a PK analysis at 60, 120 min, 6 h, 12 h after PIPAC but results at 24 hours are available. Is it a mistake?

I apologize, it is an oversight. I corrected it.

  • The PK analysis showed no or very slow plasma decrease of CDDP and OXA. In figures, the plasma concentration at 24 hours is close to concentration at 1 hour. These results are surprising and could be discussed.

Thank you for the comment. The pharmacokinetics results highlighted a very slow decrease of the plasma concentration of CDDP and OXA over time. This behavior can be related to the binding of the two platin compounds to the plasma proteins. The plasma concentration reported comprised free platinum levels and the protein-bounded portion. The discussion was modified accordingly.

  • The authors describe in the chapter « method » a drug tissue analysis with multiple biopsies. However no data are available in results.

Peritoneal samples were taken for histological confirmation of PC and drug tissue concentration evaluation: this will be the objective of a second paper.

  • The authors discuss the cumulative toxicity. It’s relevant for repeat cycle of chemotherapy. However, no clinical or pharmaokinetic data are avalaible in this study to assess this cumulative toxicity. The analysis was done after the first PIPAC and not after the others. Some data of number of PIPAC done and toxicities after second or third PIPAC could be interesting to assess the cumulative toxicity.

Thank you for the consideration. As described in the paper, patients were planned to undergo only one PIPAC procedure: therefore, cumulative toxicity cannot be evaluated. Considering that multiple retrospective and prospective study about PIPAC have shown no cumulative toxicity with standard dose, the possibility of using higher dosages would not exclude the chance of repeated treatments. Moreover, literature reported no significant accumulation in plasma ultrafiltrate after multiple dosing at 130 mg/sm every 3 weeks: it can be deduced that even after repeated PIPACs no plasma drug accumulation should be detected.  Time intervals between one PIPAC procedure and the following one can be in any case extended if necessary.

  • Another phase I study of PIPAC with oxaliplatin found a MTD of 90mg/m2, lower than 135mg/m2 of this study. The authors could discuss this different results.

PIPAC-OX reported 120 mg/m2 as recommend dose for phase 2 studies about PIPAC: at this dose level, that was the highest planned, no DLT was registered. Moreover, in the paper they planned 5 cohorts of patients (45, 60, 90, 120 and 150 mg/m2), but even if they did not established the MTD, the study was terminated after the dose cohort of 120 mg/m2.

By comparing these two studies it is clear that the toxicities recorded even with high OXA dosed are actually mild or moderate in both articles.

In the Singapore casuistry 3 cases (of which one severe, the only grade 3 complication) of pancreatitis are reported; in our series neither analogous cases nor cases of amylase and / or lipase elevation have been recorded. As reported by the authors themselves, acute pancreatitis is not a complication typically related to the administration of OXA as intraperitoneal pressurized aerosol. I therefore believe it may be accidental or perhaps due to a method of carrying out the procedure and it should still be investigated further. On the contrary, abdominal pain, which in our series of patients treated with OXA turns out to be the most frequent complication, is known to be related to the intraperitoneal administration of this drug.

In terms of radiological response, considering patients submitted to one PIPAC procedure, the Singaporean study reported a stable disease in 62.5% of patients compared to 92.3% (12/13) reported in our analysis.

The above reported considerations have been included in the paper.

Round 2

Reviewer 1 Report

The authors have addressed the comments appropriately.

The manuscript is acceptable in present form.

Author Response

Thank you very much for the review and the comments.